Three new species of free-living marine nematodes of the Microlaimus genus (Nematoda: Microlaimidae) from the continental shelf off northeastern Brazil (Atlantic Ocean)

Manoel Alex
F. Neres Patrícia
http://orcid.org/0000-0003-0921-2731 Esteves Andre M. andresteves.ufpe@gmail.com
Zoologia, Universidade Federal de Pernambuco , Recife, PE , Brazil
Venmathi Maran Balu Alagar
Electronic publication date: 2024 Apr 30
Publication date: 2024
Volume: 12
Electronic Location ID: e17355
Received 2023 Dec 7; Accepted 2024 Apr 18
Copyright: © 2024 Manoel et al.
Copyright year: 2024
Copyright holder: Silva et al.
License: This is an open access article distributed under the terms of the Creative Commons Attribution License, which permits unrestricted use, distribution, reproduction and adaptation in any medium and for any purpose provided that it is properly attributed. For attribution, the original author(s), title, publication source (PeerJ) and either DOI or URL of the article must be cited.
License URL: https://creativecommons.org/licenses/by/4.0/

Keywords: Marine nematodes, Taxonomy, Species description, South Atlantic, Nematode diversity

Funding: FACEPE Graduate Scholarship IBPG-1516-2.00/21 The Brazilian navy provided logistical support for the scientific cruise aboard the R/V Vital de Oliveira. A. Manoel has a FACEPE graduate scholarship (IBPG-1516-2.00/21). The funders had no role in study design, data collection and analysis, decision to publish, or preparation of the manuscript.

==============================
Three new species of the Microlaimus genus (Nematoda: Microlaimidae) are described from sample sediments collected in the South Atlantic, along the Continental Shelf break of Northeastern Brazil. Microlaimus paraundulatus sp. n. possesses four setiform cephalic sensillae, a buccal cavity with three small teeth, arched and slender spicules and a wave-shaped gubernaculum. Microlaimus modestus sp. n. is characterized by four small cephalic sensillae, a buccal cavity with three teeth (one large dorsal tooth), cephalated spicules and a strongly arched gubernaculum in the distal region. Microlaimus nordestinus sp. n. is characterized by the following set of features: relatively long body, eight rows of hypodermal glands that extend longitudinally along the body and a funnel-shaped gubernaculum surrounding the spicules at the distal end. An amendment of the diagnosis is proposed for the genus.

Introduction

Among the representatives of the superfamily Microlaimoidea Micoletzky, 1922, the family Microlaimidae Micoletzky, 1922 encompasses the largest number of genera and species (Tchesunov, Jeong & Lee, 2021). Amid the genera placed within Microlaimidae, records of species originally described from samples sediment of the South Atlantic are still scarce (WoRMS Editorial Board, 2024). In research carried out in Brazil in the 1950s, Gerlach described several new species of Nematoda for the hitherto “unexplored Brazilian coast”. As part of their results, four species of the genus Microlaimus de Man, 1880 were described (M. papillatus Gerlach, 1956; M. capillaris Gerlach, 1957a; M. spinosus Gerlach, 1957b and M. formosus Gerlach, 1957b) from samples collected in mangroves and sandy beaches of the southeastern coast of Brazil. Later, Aponema papillatum Pastor de Ward, 1980 and M. decoratus Pastor de Ward, 1989 were described from the Ria Deseado (Santa Cruz, Argentina). Recently, Lima, Neres & Esteves (2022) described three species of Microlaimus (M. campiensis, M. alexandri and M. vitorius) from the continental shelf of the Campos Basin, southeastern Brazil.

Microlaimus is the largest genus in the family Microlaimidae (Lima, Neres & Esteves, 2022). Nowadays, the genus includes 86 valid species (WoRMS Editorial Board, 2024). Marine representatives of this taxon are widely distributed, ranging from the intertidal zone (Leduc & Wharton, 2008) to the deepest areas of the ocean (Miljutin & Miljutina, 2009). Due to morphological similarities, species transfers between Microlaimus and other close genera belonging to the same family, have been recorded several times in the literature (Tchesunov, 2014; Leduc, 2016; Lima, Neres & Esteves, 2022). This is a result of disagreements about which morphological characters should be used to establish differences between such genera (Leduc, 2016). The main features used together to differentiate representatives of the genus Microlaimus from other genera of the same family are: annulated cuticle, with some species also presenting punctations or longitudinal bars; head often slightly set off; amphidial fovea cryptocircular or unispiral; buccal cavity small to medium-sized, armed with three teeth with an often well-developed dorsal tooth; female didelphic-amphidelphic with outstretched ovaries and male with gubernaculum without dorso-caudal apophysis (Decraemer & Smol, 2006; Techunov, 2014; Leduc, 2016; Lima, Neres & Esteves, 2022).

In the present study, representatives of the genus Microlaimus were found from samples collected in the South Atlantic, along the break of the Continental Shelf in Northeast Brazil. Here we describe the first three new species of Microlaimus for this locality. We also propose an amendment to the diagnosis of the genus.

Materials and Methods

Study area (Table 1)

The sampling process was carried out during an oceanographic campaign of the UFPE S.O.S. SEA project, in November and December 2019, on board the ship Vital de Oliveira. The sampling grid consisted of 23 collection stations arranged along the break of the Continental Shelf in Northeast Brazil, off the coast of the states of Rio Grande do Norte, Paraíba, Pernambuco, Alagoas, Sergipe, Bahia. In Table 1, information on the collection stations related to the present study are indicated. A box-corer was used to collect sediments, and the meiofauna samples were obtained with a corer (dimensions 10 cm × 10 cm).

Table 1 Collection stations, their respective coordinates and depth. The samples were collected at the break of the continental shelf in Northeast Brazil, South Atlantic.

Station	Latitude	Longitude	Depth	
2	S 05°42′54.42″	W 34°59′31.92″	60 m	
4	S 06°27′06.06″	W 34°45′53.64″	56 m	
11	S 09°15′30.54″	W 34°57′13.14″	87 m	
12	S 09°39′14.52″	W 35°15′21.66″	50 m	
13	S 09°56′55.68″	W 35°39′51.78″	44 m	
16	S 10°44′59.28″	W 36°25′32.88″	58 m	

Laboratory processing

In the laboratory, sediment samples were sieved using a 500 μm mesh followed by a 45 μm mesh sieve which was used to retain the meiobenthic organisms. The samples remaining in the 45 μm mesh were extracted with colloidal silica (Somerfield, Warwick & Moens, 2005).

Nematoda were counted (and removed) under a stereomicroscope using a Dolffus plate. All individuals were transferred to a small glass container containing a solution with 99% formaldehyde (4%) + 1% glycerin (Solution 1–De Grisse, 1969). The methodology for impregnating each animal’s body with glycerin was then applied, followed by diaphanization, according to the method described by De Grisse (1969). The individuals were mounted permanently on glass slides, as an adaptation of the method described by Cobb (1920). The genus was identified using keys provided by Warwick, Platt & Somerfield (1998) and Decraemer & Smol (2006). Species were identified through the comparison of their characteristics with those provided in the original descriptions. Drawings were made with the aid of an Olympus CX 31 optical microscope fitted with a drawing tube. Body measurements were taken using a mechanical map meter.

The holotype and one paratype (female) of each species are deposited in the Nematoda Collection of the Museum of Oceanography Prof. Petronio Alves Coelho (MOUFPE), Brazil. Other paratypes are deposited in the Meiofauna Laboratory, Zoology Department, Federal University of Pernambuco (NM LMZOO-UFPE).

The electronic version of this article in Portable Document Format (PDF) will represent a published study according to the International Commission on Zoological Nomenclature (ICZN), and hence the new names contained in the electronic version are effectively published under the Code from the electronic edition alone. This published research and the nomenclatural acts it contains have been registered in ZooBank, the online registration system for the ICZN. The ZooBank LSIDs (Life Science Identifiers) can be resolved and the associated information viewed through any standard web browser by appending the LSID to the prefix http://zoobank.org/. The LSID for this publication is: urn:lsid:zoobank.org: pub: 414C399D-A60E-494E-9E36-C4866FBC9539. The online version of this research is archived and available from the following digital repositories: PeerJ, PubMed Central and CLOCKSS.

Results

SYSTEMATICS

Class CHROMADOREA Inglis, 1983

Subclass CHROMADORIA Pearse, 1942

Order Microlaimida Leduc, Verdon & Zhao, 2018

Superfamily Microlaimoidea Micoletzky, 1922

Family Microlaimidae Micoletzky, 1922

Genus Microlaimus de Man, 1880

Syn Microlaimoides Hoeppli, 1926; Paracothonolaimus Schulz, 1932

Diagnosis. (Emended from Lima, Neres & Esteves, 2022): Cuticle transversely striated, punctuations or longitudinal bars may be present. Lateral differentiation in the form of lateral alae occurs in M. falciferus Leduc & Wharton, 2008. Cephalic region often set off. Presence or absence of association between hypodermal glands with pores or setae, small somatic setae occur in some species. Anterior sensilla arranged according to pattern 6 + 6 + 4: six inner labial setae, usually papilliform; six outer labial setae, papilliform or setiform; and four cephalic setae. Amphidial fovea cryptocircular or unispiral (=cryptospiral). Presence or absence of sexual dimorphism in amphidial fovea size. Buccal cavity small to medium-sized, armed (except in M. nympha) with two to five (three teeth in most species) small or well-developed teeth, especially the dorsal tooth. Transverse cuticularized band or ring may be present in buccal cavity. Most species have two testes extending in opposite directions; some with two anterior testes, others with only one testis, positioned anteriorly or posteriorly. Pre-cloacal supplements absent or present (papilliform, tubular, or small pores). Spicules usually short and arcuate, seldom long and slender. Gubernaculum usually present and without dorso-caudal apophysis. Female didelphic-amphidelphic, with outstretched ovaries. Tail predominantly conical.

Type species: Microlaimus globiceps de Man, 1880.

Microlaimus paraundulatus sp. n.

(Table 2; Figs. 1–2)

Table 2 Morphometric data of Microlaimus paraundulatus sp. n.

Microlaimus paraundulatus sp. n.	Holotype
male	Paratype male	Paratype female	
Body length	457.5	439	465	
Cephalic setae length	3	3	3	
Head diameter at level of the cephalic setae	7	7	7	
Cephalic setae in relation to head diameter (%)	43%	43%	43%	
Distance from anterior end to amphidial fovea	11	11	13	
Distance from anterior end to amphidial fovea in relation to head diameter	1.6	1.6	1.9	
Amphidial fovea diameter (maximum width)	4	4	5	
Body diameter at level of the amphidial fovea	9	9	9	
% of the amphidial fovea diameter in relation to corresponding body diameter	44%	44%	56%	
Pharynx length	82	83	84	
Position of nerve ring from anterior end	47	-	49.5	
Nerve ring position in relation to pharynx length (%)	57%	-	59%	
Pharyngeal bulb diameter	11	11	12	
Body diameter at level of the pharyngeal bulb	14	15	15	
% of basal bulb diameter in relation to corresponding body diameter	79%	73%	80%	
Maximum body diameter	15	16	18	
Anal or cloacal body diameter	12	13	10	
Tail length	59	55.5	55	
Length of spicule along arc	27	23	*	
Length of spicule along cord	18	16	*	
Length of gubernaculum	13	13	*	
Length of gubernaculum in relation to length of spicule along arc (%)	48%	57%	*	
Length of spicule along arc in relation to cloacal body diameter	2.3	1.8	*	
Distance from anterior end to vulva	*	*	244.5	
Position of vulva from anterior end (%)	*	*	53%	
Body diameter in vulva region	*	*	18	
Anterior ovary length	*	*	110	
Posterior ovary length	*	*	95	
Reproductive system	303	224	205	
% of reproductive system in relation to body length	66%	51%	44%	
a	30.5	27	26	
b	6	5	5.5	
c	8	8	8	
c′	5	4	6	
Note:

The measurements are expressed in micrometers, or if noted, as a percentage or ratio. Not applicable (*); not available for measurement (-); a, b, c, c′ = de Man’s (1880) ratios.

Figure 1 Microlaimus paraundulatus sp. n. holotype male and paratype female.

Holotype male: (A) overview, (B) anterior region, (C) posterior region. Paratype female: (D) overview, (E) anterior region.

Figure 2 Microlaimus paraundulatus sp. n. holotype male, paratype male and paratype female.

Holotype male: (A) anterior end (arrows indicating: cs, cephalic setae; amph, amphidial fovea), (B) anterior region (arrow indicating bulb). Paratype male: (C) anterior end (arrow indicating dorsal tooth). Holotype male: (D) spicule and gubernaculum. Paratype female: (E) anterior end (arrow indicating: amph= amphidial fovea), (F) reproductive system (arrows indicating: V, vulva; ov.a., anterior ovary; ov.p., posterior ovary).

Material studied. Holotype male (MOUFPE 0017), paratype female (MOUFPE 0017) and 1 male paratype (473 NM LMZOO-UFPE).

Type locality. South Atlantic Ocean, Continental shelf of the State of Rio Grande do Norte, Brazil, station 2 (S 05°42′54.42″W 34°59′31.92″), November 28, 2019, 60 m. Paratypes found in the same locality.

Etymology. The gubernaculum of Microlaimus paraundulatus sp. n. has a wave-shaped structure, similar to the gubernaculum of Microlaimus undulatus Gerlach, 1953.

Holotype male. Body cylindrical 457.5 μm long. Maximum body diameter corresponding to 2.1 times the head diameter. Cuticle striated posteriorly to cephalic setae insertion. Cuticular pores and somatic setae not observed. Six inner and six outer papilliform labial sensillae. Four cephalic sensillae in the shape of thin setae 3 μm long, corresponding to 43% of head diameter. Head slightly set off. Amphidial fovea cryptocircular, located 11 μm from anterior end (1.6 times the head diameter) and occupying 44% of corresponding body diameter. Buccal cavity weakly cuticularized. Cheilostoma rugae indiscernible under a light microscope. Three small teeth, difficult to see (a slightly larger dorsal tooth and two smaller ventrosublateral). Pharynx (82 μm long) with terminal oval bulb. Bulb occupying 79% of corresponding body diameter. Cardia embedded in intestine. Nerve ring situated at 57% of the pharynx length from anterior end. Ventral gland and secretory-excretory pore not observed. Reproductive system with two testes extending in opposite directions. Spicules slender and arched. Gubernaculum curved and wave-shaped. Precloacal supplements absent. Three caudal glands. Tail conical, about 4.9 times the cloacal body diameter.

Paratype female. Similar to male. Body measuring 465 μm in length, with a maximum diameter of 18 μm. Cephalic sensilla equivalent to 43% of head diameter. Amphidial fovea, occupying 56% of corresponding body width and located 13 μm from anterior end. Buccal cavity, teeth and pharynx similar to that of the male. Basal bulb occupying 80% of the corresponding body diameter. Nerve ring situated at 59% of the pharynx length, from anterior end. Vulva located 244.5 μm from anterior end, at 53% of body length. Reproductive system didelphic-amphidelphic, with outstretched ovaries. Anterior ovary situated to the right side of intestine, posterior ovary to the left side of intestine. Tail conical, about 5.5 times the anal body diameter.

Diagnosis. Microlaimus paraundulatus sp. n. characterized by its body length (439–465 μm). Cuticle finely annulated. Head slightly set off. Four cephalic sensillae in the shape of thin setae (3 μm long), corresponding to 43% of head diameter. Amphidial fovea occupying 44% of the corresponding body diameter in the males and 56% in the female, located at about 1.6–1.9 times the head diameter. Buccal cavity with three small teeth, one dorsal and two ventrosublateral, the dorsal is slightly larger. Spicule arched and slender (1.8–2.3 times the cloacal body diameter) with a wave-shaped gubernaculum. Tail conical which corresponds to 4–6 cloacal or anal body diameter.

Differential diagnosis (Table 3). Firstly, it is important to establish that only males of each species mentioned in this section were used in the comparisons with males of the new species. More detailed information, such as some measurements and proportions, about females is absent in the original descriptions of some of the species in question. The measurements and proportions missing from the descriptions were obtained from available images.

Table 3 Comparison of species Microlaimus paraundulatus sp. n. with morphologically similar species (only males).

	M. copulatus	M. limnophilus	M. porus	M. zosterae *	M. paraundulatus	
Body length	320–330	401–470	380	618–621	439–457.5	
a	18–20	22.8–29.3	21.1	27–29.4	27–30.5	
b	4.1–4.6	5.6–6.5	4.9	6.7–7.2	5–6	
c	6–7.2	7–8.8	5.4	9.4–9.9	8	
c′	3.2	3.2–4.1	5.1	3.6–3.9	4–5	
Cephalic setae in relation to head diameter (%)	17%	23–25%	38%	20%	43%	
amph%	40%	33–35%	<50%**	43–45%	44%	
Amph/hd	1.8	1.9	1.6	1.9	1.6	
Length of spicule along arc in relation to cloacal body diameter	2.7–2.8	1.3	1.7	1.6	1.8–2.3	
Length of gubernaculum in relation to length of spicule along arc (%)	30%	47%	42%	60%	48–56%	
Precloacal supplements	+	+	−	+	−	
Cuticular pores	−	−	+	−	−	
Notes:

Information (measurements/proportions) of males of the species Microlaimus that concomitantly have cephalic setae <50% of head diameter; amphidial fovea between 1.6 and 1.9 times the head diameter; amphidial fovea <50% of the head body diameter.

The measurements are expressed in micrometers, or if noted, as a percentage or ratio. Present (+) or absent (−); a, b, c, c′ = de Man’s (1880) ratios; distance of amphidial fovea from anterior end in relation to head diameter (Amph/hd); percentage of the amphidial fovea diameter in relation to corresponding body diameter (amph%).

* Based on the redescription of the species Microlaimus zosterae Allgén, 1930 provided by Kovalyev & Tchesunov (2005).

** Proportion measured from the drawing of the female paratype. Missing information for the holotype male.

Microlaimus paraundulatus sp. n. resembles M. undulatus Gerlach, 1953 mainly due to the peculiar shape of the gubernaculum (wave-shaped). Additionally, in both species the amphidial fovea occupies a similar proportion of the corresponding body diameter (44% in M. paraundulatus sp. n. and 42% in M. undulatus) and the spicules are very similar in length (23–27 μm in M. paraundulatus sp. n. and 27 μm in M. undulatus). However, the species differ in the length of the cephalic setae (3 μm in the new species vs 7 μm in M. undulatus), the position of the amphidial fovea from the anterior end (1.6 times the diameter of the head in M. paraundulatus sp. n. vs 0.7 times the diameter head diameter in M. undulatus); the shape of the spicules (thin in the new species vs robust in M. undulatus) and the ratio between the length of the spicules and the diameter of the cloaca (1.8–2.3 in M. paraundulatus sp. n. vs 1.1 in M. undulatus).

The other four species of the genus (M. copulatus Jensen, 1988; M. limnophilus Turpeenniemi, 1997; M. porus Bussau, 1993 and M. zosterae Allgén, 1930) that morphologically resemble M. paraundulatus sp. n. are included in Table 3. Males of these species share the combination of three features with the new species: length of cephalic setae less than 50% of the head diameter; amphidial fovea with anterior edge positioned between 1.6 and 1.9 times the head diameter in relation to the anterior end; amphidial fovea occupies less than 50% of the corresponding body diameter. Only the percentage of the corresponding body diameter occupied by the amphidial fovea for M. porus was obtained from the drawing of the female paratype (proportion not reported and anterior end not drawn in the original description of the male holotype; sexual dimorphism absent). Microlaimus undulatus, like the other species of the genus, does not present these characteristics simultaneously and therefore the species was not included in Table 3.

The new species differs from M. copulatus with regard to index “a” (27–30.5 in M. paraundulatus sp. n. vs 18–20 in M. copulatus), the ratio (%) of gubernaculum length in relation to spicule length along arc (48–56% in the new species vs 30% in M. copulatus) and the precloacal supplement (absent in M. paraundulatus vs a precloacal papilla in M. copulatus). Additionally, the gubernaculum has dorsal apophyses and an irregular shape in M. copulatus and is wave-shaped without apophyses in M. paraundulatus sp. n. Microlaimus paraundulatus sp. n. differs from M. limnophilus in terms of the length (%) of the cephalic setae in relation to head diameter (43% in M. paraundulatus sp. n. vs 23–25% in M. limnophilus), the shape of the gubernaculum (wave-shaped in M. paraundulatus sp. n. vs rod-like in M. limnophilus) and the ratio of the spicules in relation to the cloacal body diameter (2.3 in M. paraundulatus sp. n. vs 1.3 in M. limnophilus). Furthermore, M. limnophilus possesses two pore-like precloacal supplements, while in the new species the precloacal supplements are absent.

The new species differs from M. porus Bussau, 1993 with regard to the number of teeth (three in M. paraundulatus vs two in M. porus), the shape of the gubernaculum (wave-shaped in the new species vs simple in M. porus) and the values of the indices “a” (30.5 in the new species vs 21.1 in M. porus) and “c” (8 in the new species vs 5.4 in M. porus). Additionally, M. porus has rows of pores distributed throughout the body. This feature is absent in the new species.

Based on the redescription of the species, M. zosterae, provided by Kovalyev & Tchesunov (2005), whose original description was based on females alone, M. paraundulatus sp. n. differs from males of M. zosterae in terms of the shape of the gubernaculum (wave-shaped in the new species vs curved, tapering to the ends, wider in the middle part in M. zosterae) and the precloacal supplements that are absent in M. paraundulatus sp. n. vs present in M. zosterae.

Microlaimus modestus sp. n.

(Table 4; Figs. 3–5)

Table 4 Morphometric data of Microlaimus modestus sp. n.

Microlaimus modestus sp. n.	Holotype male	Paratype male 1	Females paratypes	
Body length	342	344	331.5–359	
Cephalic setae length	2	2	2	
Head diameter at level of the cephalic setae	9	9	8–9	
Cephalic setae in relation to head diameter (%)	22%	22%	22–25%	
Distance from anterior end to amphidial fovea	10	11	10–12	
Distance from anterior end to amphidial fovea in relation to head diameter	1.1	1.2	1.1–1.5	
Amphidial fovea diameter (maximum width)	6	6	5–6	
Body diameter at level of the amphidial fovea	10.5	11	10	
% of the amphidial fovea diameter in relation to corresponding body diameter	57%	55%	50–60%	
Pharynx length	82	82	78–85.5	
Position of nerve ring from anterior end	51	48	47–49	
Nerve ring position in relation to pharynx length (%)	62%	59%	57%–63%	
Pharyngeal bulb diameter	15	17	13–17	
Body diameter at level of the pharyngeal bulb	17	19	18–21	
% of basal bulb diameter in relation to corresponding body diameter	88%	89%	65–84%	
Position of secretory-excretory pore from anterior end	56	56	53	
Maximum body diameter	20	20	22–29.5	
Anal or cloacal body diameter	15	16	13–16	
Tail lenth	48	54	35–57	
Length of spicule along arc	34	32	*	
Length of spicule along cord	20	22	*	
Length of gubernaculum	19	16	*	
Length of gubernaculum in relation to length of spicule along arc (%)	56%	50%	*	
Length of spicule along arc in relation to cloacal body diameter	2.3	2.0	*	
Distance from anterior end to vulva	*	*	195–207	
Position of vulva from anterior end (%)	*	*	57–59%	
Body diameter in vulva region	*	*	22–29	
Anterior ovary length	*	*	49–65	
Posterior ovary length	*	*	52.5–65	
Reproductive system length	163	158	107.5–130	
% of reproductive system in relation to body length	48%	46%	31–37%	
a	17	17	12–16	
b	4	4	4.1–4.4	
c	7	6	6–9	
c′	3	3	3–4	
Note:

The measurements are expressed in micrometers, or if noted, as a percentage or ratio. Not applicable (*); a, b, c, c′ = de Man’s (1880) ratios.

Figure 3 Microlaimus modestus sp. n. holotype male and paratype female.

Holotype male: (A) overview, (B) anterior region, (C) posterior region. Paratype female: (D) overview, (E) anterior region.

Figure 4 Microlaimus modestus sp. n. holotype male and paratype male.

Holotype male: (A) anterior region, (B) anterior region (arrow indicating cephalic setae), (C) anterior region (pharynx and bulb), (D) anterior end (arrow indicating dorsal tooth), (E) posterior end (spicule and gubernaculum). Paratype male: (F) habitus (arrow indicating testicle).

Figure 5 Microlaimus modestus sp. n. paratype female and paratype female 2.

(A) Anterior region, (B) anterior region (arrow indicating cephalic setae), (C) anterior end (arrow indicating dorsal tooth). Paratype female 2: (D) habitus (ov.a., anterior ovary; ov.p., posterior ovary; V, vulva).

Material studied. Material studied. Holotype male (MOUFPE 0018), paratype female (MOUFPE 0019), one male paratype (474 NM LMZOO-UFPE) and four female paratypes (475–478 NM LMZOO-UFPE).

Type locality. South Atlantic Ocean, Continental shelf of the State of Sergipe, Brazil, station 16 (S 10°44′59.28″W 36°25′32.88″), December 09, 2019, 58 m. Paratypes found in the same locality.

Etymology. Due to its relatively small body length. Latin modestus: short in length.

Holotype male. Body cylindrical 342 μm long. Maximum body diameter corresponding to 2.2 times the head diameter. Cuticle striated posteriorly to cephalic setae insertion. Cuticular pores and somatic setae not observed. Anterior sensilla arranged in the 6 + 6 + 4 pattern: six inner labial papilliform sensilla, six outer labial papilliform sensilla and four short cephalic setae (2 μm long), corresponding to 22% of head diameter. Head slightly set off. Amphidial fovea cryptocircular, located 10 μm from anterior end and occupying 57% of corresponding body diameter. Buccal cavity cuticularized. Cheilostoma rugae indiscernible under a light microscope. Three cuticularized teeth, one large dorsal tooth and two smaller ventrosublateral teeth. Pharynx (82 μm long) with terminal oval bulb. Bulb occupies 88% of corresponding body diameter. Cardia embedded in intestine. Nerve ring situated at 62% of the pharynx length, from anterior end. Secretory-excretory pore located 56 μm from anterior end (about 68% of the pharynx length). Ventral gland not observed. Reproductive system with single anterior outstretched testis on the right side of intestine. Spicules arched, with cephalated proximal end. Gubernaculum simple with strongly arched distal region. Precloacal supplements absent. Three caudal glands. Tail conical, about 3.2 times the cloacal body diameter.

Paratype female. Similar to male. Body measuring 359 μm in length, and maximum diameter 23 μm. Cephalic sensilla equivalent to 24% of head diameter. Amphidial fovea, occupying 56% of corresponding body width and located 11 μm from anterior end. Buccal cavity, teeth and pharynx similar to that of the male. Basal bulb occupies 84% of the corresponding body diameter. Nerve ring situated at 59% of the pharynx length, from anterior end. Secretory-excretory pore located 53 μm from anterior end (about 65% of the pharynx length). Vulva located 206 μm from anterior end, at 57% of body length. Reproductive system didelphic-amphidelphic, with outstretched ovaries. Anterior ovary situated to right side of intestine, posterior ovary to left side of intestine. Anterior and posterior ovary measuring 49 and 64 μm, respectively. Tail conical, about 3.6 times the anal body diameter.

Diagnosis. Microlaimus modestus sp. n. is characterized by its small body length (331.5–359 μm). Cuticle finely annulated. Head slightly set off. Inner and outer labial setae in the shape of papillae. Four short cephalic setae (2 μm long) that correspond to 22–25% of head diameter. Amphidial fovea accounts for 50–60% of the corresponding body diameter, located at about 1.1–1.2 times the head diameter in males and 1.1–1.5 in females. Buccal cavity with three teeth, one large dorsal and two smaller ventrosublateral. Paired spicules arched, with cephalated proximal end. Gubernaculum strongly arched distal region.

Differential diagnosis (Table 5). Firstly, it is important to establish that only males of each species mentioned in this section were used for comparisons with males of the new species. This is because only males of M. acanthus (Jayasree & Warwick, 1977; Kovalyev & Tchesunov, 2005) have been described. Measurements and proportions missing from the descriptions were obtained from available images.

Table 5 Comparison of species Microlaimus modestus sp. n. with morphologically similar species (only males).

	M. acanthus	M. microseta	M. modestus	
L	845–1,175	1,205	342–344	
a	31.3–36.7	55	17	
b	7.4–8	8.6	4	
c	9.9–10.7	14.2	6–7	
c′	3.5	4.25	3	
Cephalic setae in relation to head diameter (%)	40–50%	25%	20%	
amph%	67–75%	60%	57–55%	
Amph/hd	1.1–1.8*	1.1	1.1–1.2	
Length of spicule along arc in relation to cloacal body diameter	1.1–1.4	1.3	2–2.3	
Length of gubernaculum in relation to length of spicule along arc (%)	53–60%	54%	50–56%	
Precloacal supplements	+	−	−	
Cuticular pores	−	−	−	
Notes:

Information (measurements/proportions) of males of the species Microlaimus that concomitantly have cephalic setae ≤50% of the head diameter; amphidial fovea between 1.1 and 1.5 times the head diameter; amphidial fovea >50% of head diameter.

The measurements are expressed in micrometers, or if noted, as a percentage or ratio. Present (+) or absent (−); a, b, c, c′ = de Man’s (1880) ratios; distance of amphidial fovea from anterior end in relation to head diameter (Amph/hd); percentage of the amphidial fovea diameter in relation to corresponding body diameter (amph%).

* M. acanthus showed a greater variation in the relative position of the amphidial fovea (1.1–1.8 times the head diameter). Despite this, due to other similarities with the new species, we will include it in the table for comparison purposes.

Two species of the genus (M. acanthus and M. microseta Gerlach, 1953) that morphologically resemble M. modestus sp. n. are included in Table 5. Males of these species share this combination of three features with the new species: cephalic setae with a length less than or equal to 50% of the head diameter; amphidial fovea occupying more than 50% of the corresponding body diameter; amphidial fovea with anterior edge positioned between 1.1 and 1.5 times the head diameter in relation to the anterior end. M. acanthus showed a greater variation in the relative position of the amphidial fovea (1.1–1.8 times the head diameter). Nevertheless, due to other similarities with the new species, it was included in the table for comparison purposes.

M. modestus sp. n. shares the ratio between gubernaculum length and spicule length (between 50–60% in the three species), as well as the de Man’s ratio c′ (between 3–4.25) with M. acanthus and M. microseta. However, the values of the other de Man’s ratio (a, b and c) for M. modestus sp. n. are relatively low compared to M. acanthus and M. microseta (see Table 4). Furthermore, the ratio between the length of the spicule along the arc and cloacal body diameter is higher in M. modestus sp. n. (2–2.3) when compared to the ratios observed in M. acanthus and M. microseta (between 1.1–1.4). Additionally, the new species differs from M. acanthus and M. microseta with regard to the shape of the gubernaculum (simple with strongly arched distal region in M. modestus sp n. vs pointed and narrow proximally and expanded distally in M. acanthus vs narrow and simple in M. microseta). M. acanthus presents four to six prominent precloacal supplements in the form of robust setae, while in the new species precloacal supplements are absent.

Microlaimus nordestinus sp. n.

(Table 6; Figs. 6–8)

Table 6 Morphometric data of Microlaimus nordestinus sp. n.

Microlaimus nordestinus sp. n.	Holotype male	Males
paratypes	Females paratypes	
Body length	1,450	1,273.5–1,405.5	1,080–1,404	
Cephalic setae length	6	5–6	6	
Head diameter at level of the cephalic setae	9	8–10	8	
Cephalic setae in relation to head diameter (%)	67%	56–67%	75%	
Distance from anterior end to amphidial fovea	17	14–19	17	
Distance from anterior end to amphidial fovea in relation to head diameter	1.9	1.6–2.1	2.1	
Amphidial fovea diameter (maximum width)	6	5–6	5	
Body diameter at level of the amphidial fovea	13	12–15	12–14	
% of the amphidial fovea diameter in relation to corresponding body diameter	46%	36–43%	36–42%	
Pharynx length	108	98–107	97–103	
Position of nerve ring from anterior end	66	58–68	60	
Nerve ring position in relation to pharynx length (%)	61%	54–65%	58%	
Pharyngeal bulb diameter	16	16%	16	
Body diameter at level of the pharyngeal bulb	24	23–24	25	
% of basal bulb diameter in relation to corresponding body diameter	67%	67–70%	64%	
Maximum body diameter	25	25–31	26–31	
Anal or cloacal body diameter	21	20–25	16–17	
Tail lenth	101	94–105	103.5–111	
Length of spicule along arc	31	28–31	*	
Length of spicule along cord	27	25.5–28	*	
Length of gubernaculum	14	13–17	*	
Length of gubernaculum in relation to length of spicule along arc (%)	45%	42–55%	*	
Length of spicule along arc in relation to cloacal body diameter	1.5	1.2–1.5	*	
Precloacal supplement closest to cloaca	16	13–17	*	
Precloacal supplement farthest from the cloaca	24	21–28	*	
Distance from anterior end to vulva	*	*	540–756	
Position of vulva from anterior end (%)	*	*	50–54%	
Body diameter in vulva region	*	*	26	
Anterior ovary length	*	*	130.5–246	
Posterior ovary length	*	*	132	
Reproductive system length	834	784	262.5	
% of reproductive system in relation to body length	58%	57%	24%	
a	58	42–55	41.5–45	
b	13	12.5–14	11–14	
c	14	13–15	10–13	
c′	5	4–5	6.5	
Note:

The measurements are expressed in micrometers, or if noted, as a percentage or ratio. Not applicable (*); a, b, c, c′ = de Man’s (1880) ratios.

Figure 6 Microlaimus nordestinus sp. n. holotype male and paratype female.

Holotype male: (A) overview, (B) anterior region, (C) posterior region. Paratype female: (D) overview, (E) anterior region.

Figure 7 Microlaimus nordestinus sp. n. holotype male.

(A) Anterior region (arrows indicating: cs, cephalic setae; amph, amphidial fovea), (B) anterior region (arrow indicating bulb), (C) anterior end (arrow indicating buccal cavity), (D) posterior end (spicule), (E) posterior end (gubernaculum), (F) posterior end (arrow indicating precloacal papilla), (G) tail.

Figure 8 Microlaimus nordestinus sp. n. paratype female 1 and paratype female 2.

Paratype female 1: (A) anterior region. Paratype female 2: (B) anterior region (arrow indicating cephalic setae). Paratype female 1: (C) anterior region (pharynx and bulb), (D) anterior region (arrow indicating buccal cavity), (E) tail, (F) vulva.

Material studied. Holotype male (MOUFPE 0020), paratype female (MOUFPE 0021), five male paratypes (479–483 NM LMZOO-UFPE) and one female paratype (484 NM LMZOO-UFPE).

Type locality. South Atlantic Ocean, Continental shelf of the State of Alagoas, Brazil, station 11 (S 09°15′30.54″ W 34°57′13.14″), November 26, 2019, 87 m.

Locality of paratypes. Paratype female 1: South Atlantic Ocean, Continental shelf of the State of Alagoas, Brazil, station 11 (S 09°15′30.54″ W 34°57′13.14″), November 26, 2019, 87 m. Paratype males (1–3): South Atlantic Ocean, Continental shelf of the State of Alagoas, Brazil, station 12 (S 09°39′14.52″ W 35°15′21.66″), November 25, 2019, 50 m. Paratype males (4 and 5): South Atlantic Ocean, Continental shelf of the State of Alagoas, Brazil, station 13 (S 09°56′55.68″ W 35°39′51.78″), November 25, 2019, 44 m. Paratype female 2: South Atlantic Ocean, Continental shelf of the State of Rio Grande do Norte, Brazil, station 04 (S 06°27′06.06″ W 34°45′53.64″), November 27, 2019, 56 m.

Etymology. Nordestinus is the Latinized form of the term “nordestino”. In Brazil, “nordestino” refers to something or someone originating from the northeastern region of the country.

Holotype male. Body cylindrical, 1,450 μm long. Maximum body diameter corresponding to 2.8 times head diameter. Cuticle striated posteriorly to cephalic setae insertion. Four sublateral, two subventral and two subdorsal rows of hypodermal glands that begin after the amphidial fovea and extend longitudinally along the body. Hypodermal glands visible up to about 57% of the total length of the tail from the cloaca. Cuticular pores and somatic setae not observed. Anterior sensilla arranged in the 6 + 6 + 4 pattern: six inner labial papilliform sensilla, six outer labial papilliform sensilla and four cephalic sensilla (6 μm long), corresponding to 67% of head diameter. Head slightly set off. Amphidial fovea unispiral, located 17 μm from anterior end (1.9 times head diameter) and occupying 46% of corresponding body diameter. Buccal cavity weakly cuticularized. Cheilostoma rugae indiscernible under a light microscope. Three small teeth, difficult to see (a slightly larger dorsal tooth and two smaller ventrosublateral). Pharynx (108 μm long) with terminal oval bulb. Bulb occupying 67% of corresponding body diameter. Cardia embedded in intestine. Nerve ring situated at 66% of the pharynx length, from anterior end. Ventral gland and secretory-excretory pore not observed. Reproductive system with single anterior outstretched testis on right side of intestine. Spicules arched, with proximal portion cephalized. Gubernaculum funnel-shaped surrounding the spicules at the distal end. Two precloacal papilla present. The closest is about 16 μm from the cloaca and the second at 24 μm. Three caudal glands. Tail conical with cylindrical terminal portion, 4.8 times the cloacal body diameter.

Paratype female. Similar to male. Body measuring 1,404 μm in length, with a maximum diameter of 31 μm. Rows of hypodermal glands similar to the male. Hypodermal glands visible up to about 60% of the total length of the tail from the anus. Cephalic sensilla equivalent to 75% of head diameter. Amphidial fovea occupies 36% of corresponding body width and located 17 μm from anterior end. Buccal cavity, teeth and pharynx similar to that of males. Basal bulb occupies 64% of the corresponding body diameter. Nerve ring situated at 58% of the pharynx length, from anterior end. Secretory-excretory located after the nerve ring and 74 μm from the anterior end. Ventral gland located immediately posterior to pharynx. Vulva located 756 μm from anterior end, at 54% of body length. Reproductive system didelphic-amphidelphic, with outstretched ovaries. In this paratype, the posterior ovary is apparently damaged. However, in Female paratype 1 it was possible to visualize the described pattern. Anterior ovary situated to the right side of intestine, posterior ovary to the left side of intestine. Three caudal glands. Tail conical, about 6.5 times the anal body diameter.

Diagnosis. Microlaimus nordestinus sp. n. is characterized by its long body length (1,080–1,450.5 μm). Cuticle finely annulated. Head slightly set off. Cephalic setae 5–6 μm long and corresponding to 56–75% of head diameter. Amphidial fovea occupies 36–50% of the corresponding body diameter, located at about 1.6–2.1 times the head diameter. Buccal cavity with three small teeth, one dorsal and two ventrosublateral. Four sublateral, two subventral and two subdorsal rows of hypodermal glands that begin after the amphidial fovea and extend longitudinally along the body. Hypodermal glands visible up to about a half of the total length of the tail from the cloaca. Two precloacal papilla. Gubernaculum funnel-shaped surrounding the spicules at the distal end. Tail conical with cylindrical terminal portion (4.2–6.5 times the cloacal body diameter).

Differential diagnosis. The new species shares the following features with Microlaimus cyatholaimoides de Man, 1922: anterior sensilla arrangement, where the first two are circles of papilliform setae and the third is setiform; de Man’s ratio c (9–12 in M. cyatholaimoides and 10–15 in M. nordestinus sp. n.); the presence of precloacal supplements and spicule length (33–34 μm in M. cyatholaimoides and 28–31 μm in M. nordestinus sp. n.). However, M. cyatholaimoides has a shorter total body length compared to the species described here (684–960 μm vs 1,080–1,450.5 μm in M. nordestinus sp. n.). Furthermore, the new specie differs from M. cyatholaimoides in terms of the shape of the gubernaculum (funnel-shaped surrounding the spicules at the distal end in the new specie vs lamellar in M. cyatholaimoides) and the presence of a conical tail with a cylindrical terminal portion vs conical tail in M. cyatholaimoides. Based on the illustrations provided by Man, 1922 the amphidial fovea of female M. cyatholaimoides is located 1 times the head diameter in relation to the anterior end, while in M. nordestinus sp. n. females this structure is 2.1 times the head diameter from the anterior end. Although both species have rows of hypodermic glands along the body, in M. cyatholaimoides these glands are longitudinally predominantly distributed along four sublateral rows (according to Hopper & Meyers, 1967). In M. nordestinus sp. n., the glands are distributed longitudinally along eight rows: four sublateral, two subventral and two subdorsal.

The occurrence of rows of hypodermic glands has also been reported for the species M. discolensis Bussau, 1993, M. porus Bussau, 1993, M. parviporosus Miljutin & Miljutina, 2009 and M. vitorius Lima, Neres & Esteves, 2022. For all the previously mentioned species, the occurrence of cuticular pores was also recorded. M. sergeevae Revkova, 2020 has rows of pores along the body, however the presence of rows of hypodermic glands was not mentioned. The occurrence of cuticular pores cannot be observed in any of the specimens of the new species.

Microlaimus nordestinus sp. n. differs from M. discolensis in terms of total body length (1,080–1,450.5 μm vs 425–565 μm in the latter species); external labial setae papilliform and cephalic setae setiform, while these structures are setiform and are about the same length in M. discolensis and differ with regard to de Man’s ratio c’ (4.2–6.5 in the new specie vs 2.3–3.3 in M. discolensis). Moreover, the tail is conical-shaped with a cylindrical terminal portion in the new species vs conical in M. discolensis.

The new species differs from M. porus in terms of total body length (2.4–3.8 times longer), the presence of two precloacal supplements vs absent in M. porus and the shape of the gubernaculum (funnel-shaped surrounding the spicules at the distal end in the new specie vs lamellar in M. porus). Moreover, M. nordestinus sp. n. has three small teeth vs two visible teeth in M. porus.

Microlaimus nordestinus sp. n. differs from M. parviporosus with regard to its cephalic setae which are much longer than the outer labial ones, whereas the outer labial setae and the cephalic setae are about the same length in M. parviporosus. Additionally, it differs from M. parviporosus in terms of spicule length (28–31 μm vs 16–18 μm), the shape of the gubernaculum (funnel-shaped surrounding the spicules at the distal end in the new specie vs rod-like, slightly bent anteriorly in M. parviporosus), the presence of precloacal supplements (vs absent in M. parviporosus) and body length, which is 2.6–4 times greater compared to that of M. parviporosus.

Microlaimus nordestinus sp. n. resembles M. sergeevae and M. vitorius in terms of the shape of the gubernaculum. In these species, this structure surrounds the spicule in its distal portion. Nevertheless, M. nordestinus sp. n. differs from M. sergeevae in terms of the absence of cervical setae (vs present in M. sergeevae), the tail (conical with cylindrical terminal portion without rows of setae in the new species vs conical with a slightly swollen final portion and a row of subventral setae in M. sergeevae) and the precloacal supplement (two papilla in M. nordestinus sp. n. vs eight thin channels in M. sergeevae).The new species differs from M. vitorius with regard to tail shape (conical with cylindrical terminal portion in the new species vs conical in M. vitorius), shorter spicules and gubernaculum (spicules: 28–31 μm vs 45–55 μm; gubernaculum: 13–17 μm vs 20–27 μm), the presence of three small teeth in M. nordestinus sp. n. vs. three large teeth in M. vitorius, the position of the amphidial fovea (relatively further from the anterior end in the new species compared to M. vitorius: ratio between the distance from the amphidial fovea to the anterior end and the head diameter = 1.6–2.1 in M. nordestinus sp. n. vs 0.5–0.9 in M. vitorius) and the precloacal supplement (two papilla in M. nordestinus sp. n. vs three small pores in M. vitorius).

Discussion

Although most species of the genus Microlaimus have three teeth in the buccal cavity, descriptive information on the species belonging to this genus varies with regard to this characteristic. Some species have an unarmed buccal cavity, as described for M. nympha Bussau, 1993; armed with two teeth, as described by Bussau (1993) and redescribed by Miljutin & Miljutina (2009) for M. porus; or with five teeth, as described for M. alexandri Lima, Neres & Esteves, 2022. We added variability in the number of teeth present in the buccal cavity to the diagnosis of the genus.

Specific characteristics, such as the relationship between the length of the cephalic setae and head diameter (%), the diameter of the amphidial fovea in the corresponding region of the body (%) and its position in relation to the anterior end of the body, helped to approximate M. paraundulatus sp. n. and M. modestus sp. n. to the most morphologically similar known species. The use of this combination of characters is frequently used in descriptions of Microlaimus species to express similarity relationships or to indicate differences between species (Kovalyev & Tchesunov, 2005; Gagarin & Tu, 2014; Revkova, 2020; Lima, Neres & Esteves, 2022). Taxonomic tools, such as de Man’s ratios (a, b, c and c’) and proportions between spicule length/cloacal body diameter, gubernaculum length/spicule length (%) as well as the presence and absence of cuticular pores and precloacal supplements, helped to highlight the differences between the new species and the known species that are most morphologically similar to them. Additionally, the presence of rows of hypodermic glands, such as those visualized in M. nordestinus sp. n., which may or may not be associated with pores or/and setae, can also be used as a diagnostic feature to differentiate between Microlaimus species (Jensen, 1978; Hopper & Meyers, 1967; Muthumbi & Vincx, 1999).

Although there is variation in the number of testes in Microlaimus species (one or two), this characteristic was not used to correlate the species described here with other species. In most descriptions of Microlaimus species, especially those carried out before the 2000s, information about male gonads is missing. Tchesunov (2014) reviewed the genus Aponema Jensen, 1978 and stated that the number of testes is not reported for most microlaimid species and that other characteristics are more evident and easily observable than male gonads.

Our results recorded the first three species of the Microlaimus genus described from samples collected on the Continental Shelf of Northeast Brazil. The present study increases our knowledge on the species of this taxon present in the South Atlantic and significantly expands the available knowledge of the species richness of the genus, increasing the number of Microlaimus species originally described from sediment samples collected on the coast of Brazil from seven to 10.

Additional Information and Declarations

Competing Interests

Author Contributions

Field Study Permissions

Data Availability

New Species Registration

The authors declare that they have no competing interests.

Alex Manoel performed the experiments, analyzed the data, prepared figures and/or tables, authored or reviewed drafts of the article, and approved the final draft.

Patrícia F. Neres performed the experiments, analyzed the data, authored or reviewed drafts of the article, and approved the final draft.

Andre M. Esteves conceived and designed the experiments, authored or reviewed drafts of the article, and approved the final draft.

The following information was supplied relating to field study approvals (i.e., approving body and any reference numbers):

Samples were collected by Brazilian navy that provided logistical support for the scientific cruises aboard the R/V Vital de Oliveira.

The following information was supplied regarding data availability:

All data are available in Tables 2–6.

The following information was supplied regarding the registration of a newly described species:

Publication LSID: urn:lsid:zoobank.org:pub:414C399D-A60E-494E-9E36-C4866FBC9539.

Microlaimus paraundulatus LSID: urn:lsid:zoobank.org:act:E80B83BA-5D50-4058-812F-410B01983D36.

Microlaimus modestus: LSID: urn:lsid:zoobank.org:act:223386C7-F3B0-45BF-81CE-8E52B701750A.

Microlaimus nordetinus LSID: urn:lsid:zoobank.org:act:3469C291-9173-45C8-8B28-0811E46EAB7E.

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
