# Peer review of "Three new species of free-living marine nematodes of the Microlaimus genus (Nematoda: Microlaimidae) from the continental shelf off northeastern Brazil (Atlantic Ocean)"

_PeerJ, doi:10.7717/peerj.17355_

## Round 0.1 · original submission · Major Revisions

The manuscript on "Three new species of free-living marine nematodes of Microlaimus (Nematoda: Microlaimidae) from the continental shelf off northeastern Brazil (Atlantic Ocean)" has been reviewed and the reviewers suggested major revision with their comments and suggestions. I request the authors to follow and respond to their queries carefully.

·

Basic reporting

English is generally correct and not ambiguous. The submission is self-contained and the results clear.
However, some not updated information regarding taxonomy are present, as the number of family in the superfamily or genera names (more details in the main document) and the etymology behind the names of the new species can be better clarified.

Experimental design

The experimental design is exhaustively described and easily replicable.

Validity of the findings

New features and characteristics are well and detailed described and showed in the pictures. It may be helpful to add some drawings to help to focus on the new features described.
Adding a scale in the pictures may help to get an immediate idea of the dimension of the features described.

Additional comments

This work contributes to enhance our understanding of nematode diversity, which is extremely underestimated worldwide. The work is well organized but some important taxonomical information need to be updated. Finally, fixing minor mistakes in the paper can be helpful in refining on the work.

·

Basic reporting

The manuscript is devoted to the description of three new to science species of Microlaimus. The studies are done using standard methods of zoological studies - light microscopy. In present times such studies look quite old-fashioned, molecular and electron microscopical studies a recommended.
Although manuscript is written in English it needs deep correction especially in description part.

Experimental design

Extending our knowledge in biodiversity of the planet is one of the main goals of modern zoology.
The main problem of the manuscript is the English. It should be corrected not only by a fluent speaker, but someone from the field of nematology.
I understand that it is not always possible, but some additional methods on investigation such as electron microscopy and molecular phylogeny are highly recommended in modern studies.

Validity of the findings

All findings described in the manuscript are new to science.

Additional comments

I recommend to combine all drawings of one species (male and female) in one plate. There is no need to provide separate plated for male and females, since there are many free space at the pictures.
There is one major contradiction in the genus diagnosis - gubernaculum without dorso-caudal apophysis and the description of M. copulatus that possess the dorsal apophyses.
I don't clear understand the idea of authors in choosing the species for comparison. I believe that first of all the main structures such as reproductive system or copulatory apparatus must be used. The length of cephalic setae may be used as additional feature.
For example, there are two well distinguished group of species - one with two testes and another one with single testis. Then, there are species with simple gubernaculum and species with funnel shaped gubernaculum. Etc.
Probable, if authors have tried to analysed genus and formulated some major systematic principles for this genus, it would improve the value of the manuscript.
Other small corrections are mentioned in the manuscript.

---

## Round 0.2 · Minor Revisions

The authors have revised the manuscript well; however, a reviewer still addressed some minor issues to improve the manuscript. Please follow those corrections, revise, and resubmit with your rebuttal.

·

Basic reporting

Basic reporting is clear and sufficiently referenced. Some figures can have more details, like highlighting the bulbs in Figure 2B and Figure 7B. Figures 2 A/E and 7A aren't focused on the cs, making it hard to clearly see those. The scales are sometimes imprecise: eg. the male of Microlaimus paraundulatus is reported to be 457.5 μm long, but in Figure 1, compared to the scale set as 50 micrometers, it seems to be much longer.
Minor corrections:
In introduction, 19th line: interdial -->intertidal
M. microseta etymology: latinized-->Latinized
Discussion, The present study increases our knowledge of the species --> The present study increases our knowledge of the species

Experimental design

The experimental design is clearly described, and many useful details were added compared to the first version.

Validity of the findings

This research increases the knowledge of the genus Microlaimidae in the Continental Shelf of Northeast Brazil.

·

Basic reporting

Authors have made most of the corrections. Manuscript can be published.

Experimental design

n/a

Validity of the findings

n/a

Additional comments

n/a

---

## Round 0.3 · Minor Revisions

I went through the draft, I think it has some mistakes in the taxonomical parts, so I have given my doubts in track mode of this attached file. Please check, clarify those queries and follow the WoRMS for the taxonomy. Revise and resubmit, please. Use WoRMS citations as per the Taxonomic edit history in WoRMS.

---

## Round 0.4 · accepted · Accept

Dear Andre, I am pleased to inform you that your paper has been accepted for publication since your revised version of the manuscript follows well with the peer reviewers` comments. Congratulations.